# Temperature/pH-Responsive Carboxymethyl Cellulose/Poly (*N*-isopropyl acrylamide) Interpenetrating Polymer Network Aerogels for Drug Delivery Systems

**DOI:** 10.3390/polym14081578

**Published:** 2022-04-13

**Authors:** Zhongming Liu, Sufeng Zhang, Chao Gao, Xia Meng, Shoujuan Wang, Fangong Kong

**Affiliations:** 1State Key Laboratory of Biobased Material and Green Papermaking, Key Laboratory of Pulp and Paper Science & Technology of Ministry of Education/Shandong Province, Qilu University of Technology (Shandong Academy of Sciences), Jinan 250353, China; liuzhongming126@126.com (Z.L.); gaochao0326@163.com (C.G.); mengxia7184@126.com (X.M.); nancy5921@163.com (S.W.); 2Shaanxi Provincial Key Laboratory of Papermaking Technology and Specialty Paper Development, National Demonstration Center for Experimental Light Chemistry Engineering Education, Key Laboratory of Paper Based Functional Materials of China National Light Industry, Shaanxi University of Science and Technology, Xi’an 710021, China

**Keywords:** temperature/pH-responsive, CMC/Ca^2+^/PNIPAM aerogels, drug-loaded aerogels, drug release article

## Abstract

Temperature/pH-responsive carboxymethyl cellulose/poly (*N*-isopropyl acrylamide) interpenetrating polymer network (IPN) aerogels (CMC/Ca^2+^/PNIPAM aerogels) were developed as a novel drug delivery system. The aerogel has a highly open network structure with a porosity of more than 90%, which provides convenient conditions for drug release. The morphology and structure of the CMC/Ca^2+^/PNIPAM aerogels were characterized via scanning electron microscopy (SEM), Micro-CT, X-ray photoelectron spectroscopy (XPS), pore size analysis, and cytotoxicity analysis. The analysis results demonstrate that the aerogel is non-toxic and has more active sites, temperatures, and pH response performances. The anticancer drug 5-fluorouracil (5-FU) was successfully loaded into aerogels through physical entrapment and hydrogen bonding. The drug loading and sustained-release model of aerogels are used to fit the drug loading and sustained-release curve, revealing the drug loading and sustained-release mechanism, and providing a theoretical basis for the efficient drug loading and sustained release.

## 1. Introduction

Oral administration is currently the most common administration route, and it has greatly contributed to disease treatment due to its simplicity and low production costs [1]. Recent studies have focused on the pharmacokinetic and pharmacodynamic properties of drugs, such as drug release control and biocompatibility [2]. Carboxymethyl cellulose (CMC) is hydrophilic because of its carboxylate groups, which are responsible for its properties such as bioadhesion, biocompatibility, sensitivity to environmental stimuli, and controlled drug release [3,4]. At present, some researchers use CMC to prepare oral film-coated drugs, which have great potential for the prevention and treatment of many diseases [5].

Aerogels have received increasing attention as drug carriers due to their ultra-light density, ultra-large pore volume, and outstanding high surface area [6,7]. Environment-sensitive materials with characteristics such as temperature or pH sensitivity have attracted great interest in the field of drug delivery systems [8,9]. Environmentally responsive aerogels can exhibit improved functions, such as increased drug loading capacity and better control of the drug release efficiency [10]. To obtain aerogels with dual temperature/pH response, two polymers must be used in combination. Compared with blending, interpenetrating polymer network (IPN) technology is preferable because IPN aerogels are more stable than single crosslinked networks [11,12]. The IPN technique is a new technology that exhibits superior performance, and its development can lead to the reduction of side effects and the prolongation of drug efficacy [13].

Much effort has been devoted to aerogel synthesis based on CMC and poly(*N*-isopropylacrylamide) (PNIPAM). Wang et al. successfully prepared a hybrid aerogel of chitosan, CMC, and graphene oxide for pH-controlled drug delivery [14]. Moreover, the dual-responsive aerogel was successfully prepared from the temperature/pH-sensitive graft copolymer alginate-g-P(NIPAM-co-NHMAM) for controlled drug release [15]. However, studies on the combination of CMC and PNIPAM by the IPN technique are still lacking.

The objective of this paper was to develop a novel cellulose-based aerogel as a drug delivery system. A temperature/pH responsive CMC/poly(*N*-isopropylacrylamide) IPN gel was synthesized, and the morphological structure and stimulus-responsive properties of the CMC/Ca^2+^/PNIPAM aerogel were characterized using various analytical techniques. The drug loading and sustained-release properties of CMC/Ca^2+^/PNIPAM aerogels were evaluated under different stimulation conditions. The drug loading and sustained release model was used to fit the adsorption and sustained-release curves of the aerogel, revealing the related mechanism of drug loading and sustained release.

## 2. Materials and Methods

### 2.1. Materials

Carboxymethyl cellulose (M.W:250000, DS = 1.2, 400–800 mPa·s), *N*-isopropyl acrylamide (NIPAM), *N*, *N*-methylenebisacrylamide (MBA), and *N*, *N*, *N*′, *N*′-tetramethylethylenediamine (TEMED) as an accelerator were purchased from Sigma-Aldrich, Shanghai, China. The anticancer drug 5-fluorouracil (5-FU) and phosphate-buffered saline (PBS) solution (composition: Na_2_HPO_4_, KH_2_PO_4_, NaCl and KCl) were purchased from Macklin, Shanghai, China.

### 2.2. Synthesis of Temperature-Responsive PNIPAM Aerogels

First, 2 g of NIPAM and 0.4 g MBA were dissolved in 40 mL of distilled water at 30 °C and stirred for 30 min. The solution was deoxygenated under nitrogen atmosphere for 30 min at 30 °C; 0.08 g of potassium persulfate as an initiator and 40 µL TEMED as an accelerator were added to solution. The gel solution was stirred vigorously for 3 min under nitrogen protection and then poured into a mold and sonicated for 10 min to form gel products, then continued at room temperature for 24 h. The hydrogel product was washed with a volume fraction of 50% ethanol solution, and then freeze-dried at −60 °C for 48 h to obtain the temperature-responsive PNIPAM aerogel.

### 2.3. Synthesis of Temperature/pH-Responsive CMC/Poly (N-isopropyl acrylamide) Semi-IPN Aerogels

First, 0.8 g CMC, 2 g NIPAM, and 0.4 g MBA were dissolved in 40 mL of distilled water at 30 °C and stirred for 30 min. The other reaction conditions are consistent with the thermosensitive PNIPAM aerogel, and finally the obtained the temperature/pH-responsive CMC/poly (*N*-isopropyl acrylamide) semi-IPN aerogels (CMC/PNIPAM).

### 2.4. Synthesis of Temperature/pH-Responsive CMC/Poly (N-isopropyl acrylamide) IPN Aerogels

According to the synthesis method of temperature/pH responsive CMC/poly(*N*-isopropylacrylamide) semi-IPN aerogel, 0.10 g calcium chloride was added to the reaction system as the crosslinking agent of CMC during the preparation process, and finally temperature/pH-responsive CMC/poly (*N*-isopropyl acrylamide) IPN aerogels (CMC/Ca^2+^/PNIPAM) are obtained by gel formation and freeze-drying under the same conditions.

### 2.5. Analytical Methods

#### 2.5.1. FTIR Analysis

The CMC and the PNIPAM, CMC/PNIPAM, and CMC/Ca^2+^/PNIPAM aerogels were analyzed via Fourier-transform infrared (FTIR) spectroscopy (Bruker VERTEX 70, Karlsruhe, Germany). 

#### 2.5.2. Thermal Analysis

The thermal analyses of CMC and the PNIPAM, CMC/PNIPAM, and CMC/Ca^2+^/PNIPAM aerogel samples were performed using a synchronous thermal analyzer (STA449 F3, Berlin, Germany) at 10 °C/min under the nitrogen environment.

#### 2.5.3. X-ray Photoelectron Spectroscopy Analysis

X-ray photoelectron spectroscopy (XPS) of CMC and the PNIPAM, CMC/PNIPAM, and CMC/Ca^2+^/PNIPAM aerogel samples were recorded on an EscaLab spectrometer (EscaLab Xi+, London, UK) with a width of approximately 500 µm.

#### 2.5.4. Pore Size Distribution and Cytotoxicity Analysis

Automatic Mercury Porosimeter (AutoPore V 9600, Mack, New York, USA) was used to test the pore size distribution and porosities of PNIPAM, CMC/PNIPAM and CMC/Ca^2+^/PNIPAM aerogel samples with the pressure test range from 0.10 to 61000.00 psia. 

The samples were autoclaved (121 °C, 20 min) and prepared with cell culture medium to different concentrations (5 μg/mL, 10 μg/mL, 15 μg/mL, and 20 μg/mL), co-cultured with mouse embryonic fibroblasts (NIH3T3) for 24 h, which was used to test the cytotoxicity of the aerogel samples. Each sample was tested three times and the average value was taken as the experimental data.

#### 2.5.5. Scanning Electron Microscopy and Micro-CT Analysis

The surface structures of CMC and the PNIPAM, CMC/PNIPAM, and CMC/Ca^2+^/PNIPAM aerogel samples were observed via scanning electron microscopy (SEM, Quanta 200, New Castle, DE, USA). Micro computed tomography (Micro-CT) was performed on obtained CMC/Ca^2+^/PNIPAM aerogel samples using a SkeyScan 2211 (Bruker, Rheinstetten, Germany) at 40 kV and 400 mA.

#### 2.5.6. Elemental Analysis

The elemental analyses of CMC and the PNIPAM, CMC/PNIPAM, and CMC/Ca^2+^/PNIPAM aerogel samples were performed on an elemental analyzer (Vario EL III, Elementar Analysen Systeme, Rheinstetten, Germany).

#### 2.5.7. Charge Density Analysis

Approximately 0.02 g of CMC and the PNIPAM, CMC/PNIPAM, and CMC/Ca^2+^/PNIPAM aerogel samples were dissolved in 100 mL of deionized water and ultrasonically dispersed at 30 °C for 1 h. The charge densities of the samples were measured using a particle-charge detector (Mutek, PCD 04, Dortmund, Germany) with polydiallyldimethylammonium chloride standard solution. Each sample was tested three times and the average value was taken as the experimental data.

#### 2.5.8. Swelling Behavior Analysis

The equilibrium swelling behaviors of PNIPAM, CMC/PNIPAM, and CMC/Ca^2+^/PNIPAM aerogels were measured in phosphate-buffered saline (PBS) solution under different conditions. The PNIPAM, CMC/PNIPAM, and CMC/Ca^2+^/PNIPAM aerogels were immersed in 50 mL PBS solution for 24 h. After soaking, the surface water was removed from the equilibrated PNIPAM, CMC/PNIPAM, and CMC/Ca^2+^/PNIPAM aerogels. The Equilibrium Swell Ratio (ESR) was calculated according to Equation (1):(1)$$\mathrm{ESR}=\left(W_{1}-W_{0}\right)/W_{0}$$

Here, *W*_0_ (g) and *W*_1_ (g) are the weights of the aerogel samples before and after swelling, respectively. The equilibrium swelling ratios were tested three times, and the average value was taken as the experimental data.

#### 2.5.9. Drug Loading Performance Analysis

The PNIPAM, CMC/PNIPAM, and CMC/Ca^2+^/PNIPAM aerogels were loaded with 5-FU by the static adsorption method. The aerogel was added to the 5-FU aqueous solution, and adsorbed drug at ambient temperature for 24 h. The effect of the 5-FU initial concentration (500–3000 mg/L) on the adsorption performance in an environment of pH 3 for a duration of 24 h was studied. The drug loading performance of the aerogels can be determined by the initial and final concentrations of the drug solution. Each sample was tested three times and the average value was taken as the experimental data.
(2)$$\mathrm{LE}=M_{d}/M_{s}\times 100\%$$

Here, LE is the drug loading efficiency for 5-FU, *M_d_* is the mass (mg) of the drug loaded for the aerogel, and *M_s_* is the mass for aerogel samples.

The adsorption isotherms describe the interactions between aerogels and drug molecules, which are important for optimizing drug loading and revealing loading mechanisms. The equilibrium adsorption capacity is obtained using Equation (3) and is fitted with the Langmuir model (Equation (4)) and Freundlich model (5) [16].
(3)$$q_{e}=\frac{\left(Co-Ce\right)\times V}{m}$$
(4)$$q_{e}=\frac{q_{m}K_{L}C_{e}}{1+K_{L}C_{e}}$$
(5)$$q_{e}=K_{F}C_{e}^{1/n}$$

Here, *Co* is the initial concentration of the drug, *C_e_* is the equilibrium concentration of the drug, *V* is the volume of the drug solution, m is the dry mass of the aerogels, *q_e_* is the equilibrium adsorption capacity, and *q_m_* is the maximum adsorption capacity of the aerogels. Moreover, *K_L_* and *K_F_* are the important parameters of the Langmuir and Freundlich isotherms, respectively, and n is the parameter related to the adsorption intensity in the Freundlich isotherms [17].

#### 2.5.10. Drug Release Performance Analysis

The 5-FU-loaded aerogel was measured for 5-FU release PBS buffers with different pH (3.0 and 7.4) at different temperatures (25 °C and 37 °C). A volume of drug release solution was removed and replaced with an equal volume of fresh PBS. The drug release solution was extracted periodically to calculate the cumulative percent release:Cumulative release = ((C_1_V_1_ + C_2_V_1_ + … + C_n−1_V_1_) + C_n_V_2_)/M_d_ × 100%(6)

Here, C_n_ is the 5-FU concentration at each time, V_1_ is the volume taken out each time, and V_2_ is the solution volume. Each sample was tested three times and the average value was taken as the experimental data.

The drug release profiles of the 5-FU-loaded aerogels were fitted by three mathematical models, including the Higuchi (Equation (7)) [18], first-order (Equation (8)), and Korsmeyer–Peppas model (Equation (9)) [19] to reveal the relevant release mechanism:(7)$$Q=a\times t^{1/2}$$
(8)$$Q=Q_{max}\times \left(1-e^{-kt}\right)$$
(9)$$Q=k\times t^{n}$$

Here, Q is the percentage of drug released at time *t*. In Equation (7), a is the Higuchi dissolution constant. In Equation (8), Q*_max_* and *k* is the maximum percent drug released and kinetic coefficient, respectively. In Equation (9), *k* and *n* are the release rate constant and release index, respectively.

## 3. Results

### 3.1. FTIR Analysis

The pH-responsive CMC solution was crosslinked by adding Ca^2+^. In an aqueous solution, CMC was chelated by the high-valence cation Ca^2+^ to form a stable gel structure [20]. The formation process of the pH-responsive CMC gel network is shown in Figure 1a. The polymer NIPAM was used as the temperature-sensitive compound, MBA was used as a crosslinking agent, APS was used as an initiator, and TEMED was used as an accelerator. The temperature-responsive PNIPAM hydrogel network was prepared via radical polymerization, as shown in Figure 1b [9]. The pH-responsive CMC gel network and the temperature-responsive PNIPAM hydrogel network were interspersed to construct a hydrogel with mutual transmission network, and then the solvent solution was replaced by ethanol solution, and finally freeze-dried to obtain temperature/pH-responsive CMC/poly (*N*-isopropyl acrylamide) IPN aerogels.

The FTIR spectra of CMC and the PNIPAM, CMC/PNIPAM, and CMC/Ca^2+^/PNIPAM aerogel samples are presented in Figure 1c. The broad absorbance peak at 3433 cm^−1^ corresponds to O-H stretching vibrations for CMC and aerogel samples [21], and the absorption peak at 2937 cm^−1^ corresponds to the stretching vibration of C-H for CMC and aerogel samples [21]. For the CMC curve, the characteristic peaks at 1615 and 1425 cm^−1^ correspond to the asymmetric and symmetric stretching vibrations of COO- [14]. The absorption peak at 900–1200 cm^−1^ is related to the cellulose molecular skeleton [22]. For the PNIPAM curve, the absorption peaks at 3303 cm^−1^, 3074/2981/2937 cm^−1^, 1659 cm^−1^, and 1462 cm^−1^ can be attributed to the stretching vibration of NH, the stretching vibration of CH_2_/CH_3_, the stretching vibration of C=O, and the bending vibration of NH, respectively, which are the characteristic peaks of PNIPAM polymers [23,24].

Peaks of CMC and PNIAPM polymers are present in both the CMC/PNIPAM and CMC/Ca^2+^/PNIPAM aerogels, confirming that the two polymers were blended by physical crosslinking [25]. In the CMC/Ca^2+^/PNIPAM aerogels spectrum, the peak intensity of CMC was significantly enhanced after Ca^2+^ ion crosslinking. The above results indicate that the temperature/pH-responsive CMC/Ca^2+^/PNIPAM interpenetrating network aerogel was successfully prepared.

### 3.2. Thermogravimetric Analysis and XPS Analysis

Figure 2a shows the thermogravimetric loss curves of CMC and the PNIPAM, CMC/PNIPAM, and CMC/Ca^2+^/PNIPAM aerogels. The weights of the CMC and aerogel samples gradually decreased from room temperature to 150 °C, which is attributed to the loss of moisture [26]. The main degradation temperature of CMC/NIPAM and CMC/Ca^2+^/PNIPAM aerogel was 250 °C–600 °C, which can be divided into two stages: The first stage is 250 °C–320 °C, in which the main chain of cellulose is broken [27], and the second stage is 320 °C–600 °C, which may be due to the decomposition of the PNIPAM side chain, consistent with the description in the literature [28]. The thermal weight loss curve of the CMC/PNIPAM aerogel material is basically the same as that of the PNIPAM aerogel material. For the CMC/PNIPAM and PNIPAM aerogel materials, the maximum degradation temperatures of the first stage were 295 °C and 300 °C, respectively; the maximum degradation temperatures of the second stage were 365 °C and 390 °C, respectively; and the residual amount at 600 °C was basically the same. After the CMC/Ca^2+^/PNIPAM aerogel was crosslinked by Ca^2+^ ions, the maximum degradation temperature and the residual rate of the sample increased. The crosslinking of Ca^2+^ led to a higher thermal stability of the aerogels, which was consistent with the results of the literature [29].

Figure 2b shows the XPS analysis curve of CMC and the PNIPAM, CMC/PNIPAM, and CMC/Ca^2+^/PNIPAM aerogels. The XPS spectrum of CMC does not show any signal in the N1S region, indicating that nitrogen was not present in CMC. In contrast, the PNIPAM aerogels had a high N content up to 11.73%. Moreover, the spectra of both the CMC/PNIPAM and CMC/Ca^2+^/PNIPAM aerogels had strong peaks at 399 eV, indicating that these two aerogels had higher N contents [30]. The abundant amino groups in the aerogel material can enhance the drug-loading performance of the material [31]. The CMC/Ca^2+^/PNIPAM aerogels material showed a peak of Ca element at 346 eV, indicating that Ca^2+^ played a certain crosslinking role in the aerogels’ preparation, which further confirms the successful preparation of the aerogels.

Figure 2c shows the pore size distribution curves of PNIPAM, CMC/PNIPAM, and CMC/Ca^2+^/PNIPAM aerogel materials. PNIPAM aerogel has a large wave peak between 3–50 μm, the peak value appears near 10 μm, and the main pore size distribution range is 5–20 μm. The peaks of CMC/PNIPAM and CMC/Ca^2+^/PNIPAM aerogels appear around 40 μm and 30 μm, respectively, and the main pore size distribution ranges from 10 to 50 μm. It can also be observed from the above figure that the interpenetrating network structure formed by the addition of Ca ions effectively regulates the pore structure of the aerogel, reduces the number of super-large pores, and makes the pore structure of the aerogel more uniform.

The compatibility of PNIPAM, CMC/PNIPAM, and CMC/Ca^2+^/PNIPAM to NIH3T3 cells was investigated by the MTT method. Figure 2d shows the cell survival rate of PNIPAM, CMC/PNIPAM, and CMC/Ca^2+^/PNIPAM aerogels at different concentrations. When the concentration of the three aerogel materials is in the range of 0 to 20 μg/mL, the survival rate of NIH3T3 cells is still above 90% after culturing for 24 h. Therefore, it can be considered that the aerogel drug carrier itself has no obvious toxicity; therefore, the material exhibits excellent cell compatibility. The good biocompatibility of the aerogel provides the necessary basic conditions for the next step as a drug carrier.

### 3.3. SEM and Micro CT Analysis

Figure 3 displays the SEM images of CMC (a) and the PNIPAM (b), CMC/PNIPAM (c), and CMC/Ca^2+^/PNIPAM aerogels (d). The SEM spectrum of pure CMC (Figure 3a) shows that the CMC is cylindrical and that no chemical crosslinking occurs between the fibers. Figure 3b shows that the pure PNIPAM aerogels exhibited an interrelated layered porous structure with a pore size from a few hundred nanometers to tens of microns. Figure 3c shows that CMC was well embedded in the network structure of the CMC/PNIPAM aerogels, indicating that the functional groups on the CMC surface combined with NIPAM groups to form hydrogen bonds. This causes the aerogels to form a semi-interpenetrating network structure. The SEM spectrum of the CMC/Ca^2+^/PNIPAM aerogels (Figure 3d) shows that the network formed by Ca^2+^ crosslinked with CMC, and the PNIPAM network interpenetrated to form an interpenetrating three-dimensional network structure, which formed a more uniform pore structure. The pixel size of each image is 4032 × 4032, and the resolution is 500 nm (Figure 3e,f). Micro-CT stereogram (Figure 3e) and cross section (Figure 3f) show that CMC/Ca^2+^/PNIPAM aerogel has a more uniform pore structure, consistent with SEM results.

### 3.4. Properties of CMC and Aerogels

Table 1 presents the properties of CMC and the PNIPAM, CMC/PNIPAM, and CMC/Ca^2+^/PNIPAM aerogels. As observed in the table, the CMC did not contain the N element, and the three aerogels had higher N contents. The N content of the pure PNIPAM aerogel was as high as 12.34%, and the other two aerogels had lower N contents due to the interconnection between CMC and PNIPAM. Moreover, the CMC/Ca^2+^/PNIPAM aerogels had the minimum N content (6.19%), which was mainly because the increase in Ca caused a relative decrease in N content in the aerogels. Moreover, the table shows that the porosities of the three aerogels were above 90%. The aerogels’ density gradually decreased with the increase in porosity. The CMC had a higher carboxyl functional group content, with a large amount of negative charge, and its charge density was −4.96 mmol/g. The PNIPAM aerogels with a large number of amino-functional groups had a weak charge density. The CMC/PNIPAM aerogels contained a large number of carboxyl functional groups and had a charge density of −1.08 mmol/g, while the CMC/Ca^2+^/PNIPAM aerogels had a larger amount of charge than the CMC/PNIPAM aerogels, mainly due to Ca^2+^ crosslinking, which led to an increase in the carboxyl functional group content in the aerogels.

### 3.5. Swelling Behavior Analysis

Figure 4a shows the swelling properties of PNIPAM, CMC/PNIPAM, and CMC/Ca^2+^/PNIPAM aerogels. As can be observed from the figure, as the temperature increased, the swelling properties of the three aerogel materials gradually decreased and tended to be stable. As the lower critical solution temperature (LCST) of the PNIPAM aerogel material was about 32 °C [11], the aerogels’ swelling performance decreased most at 31 °C–34 °C. The above figure also shows that the LCSTs of the CMC/PNIPAM and CMC/Ca^2+^/PNIPAM aerogels were about 37 °C, and the aerogel swelling performance was significantly reduced at this temperature. The swelling properties of the three aerogel materials changed with temperature, indicating that they had good temperature-sensitive properties. Moreover, the swelling properties of CMC/PNIPAM and CMC/Ca^2+^/PNIPAM aerogels at pH 3 were significantly lower than that at pH 7.4, which was because at lower pH conditions, CMC is protonated to insoluble CMC acid, resulting in a lower swelling ratio. Under higher pH conditions, when CMC is converted into sodium salt, its swelling behavior increases. Under the conditions of lower than LCST and high pH, the amide group of PNIPAN and the carboxyl group of CMC polymers in the aerogel form hydrogen bonds with water molecules and are in a hydrophilic state, and the swelling performance of the aerogel is high [32]. In contrast, at conditions above the LCST and at low pH, the hydrogen bonds between the polymer and water molecules are disrupted, and the conformation of the aerogel polymer changes from a loose coil to a folded chain, which is characteristic of the hydrophobic state and more low swelling properties [33,34]. Furthermore, the swelling performance of the CMC/Ca^2+^/PNIPAM aerogel was lower than that of the CMC/PNIPAM aerogel; this was mainly due to the physical crosslinking of Ca^2+^ and the carboxyl functional groups of CMC to form a tighter network structure, which led to a decrease in the CMC/Ca^2+^/PNIPAM aerogel swelling performance [35]. The LCST temperature of aerogels in acidic PBS buffer solutions is slightly lower than that in neutral PBS buffer conditions. The introduction of calcium ions leads to a slight decrease in the LCST temperature of the aerogel, which is consistent with the effect of salt ions on the critical temperature in the literature [36].

### 3.6. Drug-Loading Performance Analysis

Figure 4b shows the isothermal adsorption curves of the PNIPAM, CMC/PNIPAM, and CMC/Ca^2+^/PNIPAM aerogels. The important parameters of the fitting curve are presented in Table 2. Compared with Freundlich isotherms, the Langmuir isotherms of the three aerogels can better describe the adsorption behavior, and the correlation coefficient R^2^ is greater than 0.95, indicating that the adsorption of 5-FU on the aerogels’ surface is a single-layer adsorption [37]. Furthermore, 1/n is the heterogeneity parameter of the adsorption data; the smaller the value, the greater the heterogeneity. When 1/n = 1, the expression depicts a linear adsorption isotherm. When n is between 1 and 10, it is an easy adsorption process; when n is less than 1, the adsorption process is not easy [17]. In the Freundlich isotherm, 1/n is less than 1, indicating that the aerogels can easily adsorb drugs. The maximum adsorption capacities of the PNIPAM, CMC/PNIPAM, and CMC/Ca^2+^/PNIPAM aerogels for 5-FU were 87.24 mg/g, 161.33 mg/g, and 199.30 mg/g from Langmuir isotherms, respectively. This proves that the three aerogels can be used as effective drug carriers.

### 3.7. Drug-Release Properties Analysis

#### 3.7.1. Effect of Temperature on Drug Release

Figure 5a shows the drug release curves of drug-loaded PNIPAM, CMC/PNIPAM, and CMC/Ca^2+^/PNIPAM aerogels at 25 °C and 37 °C. For the drug-loaded PNIPAM, CMC/PNIPAM, and CMC/Ca^2+^/PNIPAM aerogels, the cumulative drug releases at 25 °C were 61%, 56%, and 52%, respectively, and those at 37 °C were 82%, 76%, and 68%, respectively. The results demonstrate that the drug release rates of the three aerogels were significantly faster at 37 °C, which may be related to the LCST for PNIPAM [11,38]. The PNIPAM polymer chain conformation expanded at 25 °C, but its expansion behavior was limited by the aerogels’ structure. This resulted in the formation of an internal hydrophobic microenvironment in aerogels, and 5-FU drugs accumulated inside the aerogels, which resulted in a low diffusion rate. When the temperature is 37 °C, the PNIPAM polymer chains will entangle and polymerize, causing the aerogels’ structure to shrink, which will accelerate the drug release from the center of the aerogels [15]. The cumulative release of the three aerogels’ drugs gradually decreased mainly due to the introduction of CMC and Ca^2+^. The introduction of CMC and Ca^2+^ can cause the aerogels to form more hydrogen bonds and other noncovalent bond forces with the drug molecules, resulting in the reduction of the drug release rate [14]. The above result demonstrates that the PNIPAM, CMC/PNIPAM, and CMC/Ca^2+^/PNIPAM aerogels have good temperature-response performance and drug release properties.

#### 3.7.2. Effect of pH on Drug Release

Figure 5b shows the drug release curves of PNIPAM, CMC/PNIPAM, and CMC/Ca^2+^/PNIPAM drug-loaded aerogels at pH 3 and pH 7.4. For the drug-loaded PNIPAM, CMC/PNIPAM, and CMC/Ca^2+^/PNIPAM aerogels at 25 °C, the cumulative drug releases at pH 7.4 were 61%, 56%, and 52%, respectively, and those at pH 3 were 58%, 90%, and 94%, respectively. The results demonstrate that the drug release rates of the CMC/PNIPAM and CMC/Ca^2+^/PNIPAM aerogels were significantly higher at pH 3 than at pH 7.4, which was due to the protonation of the carboxyl group on the drug carrier in a low pH environment; more hydrogen bonds are formed in the drug carrier, the aerogels shrinks, and the drug is consequently released by extrusion [9,11]. The above result shows that the CMC/PNIPAM and CMC/Ca^2+^/PNIPAM aerogels have good pH response performance and drug release properties.

#### 3.7.3. Drug Release Mechanism

Aerogel was chosen as a drug release carrier to study its responsive release behavior in different environments and to discuss the mechanism of its sustained-release behavior. The anticancer drug 5-FU was selected as the model drug for the sustained-release system.

The pKa of 5-FU is about 8, which makes most of its drug molecules uncharged, and the loading of aerogel drugs is mainly through the amino group in PNIPAM and the carboxyl group in CMC polymer to form hydrogen bonds with drug molecules or through the physical encapsulation of the gel network. Under different temperature and pH conditions, the aerogel network and the strength of hydrogen bonds formed with drug molecules will change differently, resulting in rapid or slow release of the drug. Figure 6 depicts the release mechanism of the drug-loaded aerogels in different solutions. As shown in Figure 6a, the PNIPAM polymer chain conformation expanded below the LCST, but its expansion behavior was limited by the aerogels’ structure. This led to the formation of an internal hydrophobic microenvironment in aerogels, and 5-FU drugs accumulated inside the aerogels, resulting in a lower diffusion rate. However, when the temperature is 37 °C, the PNIPAM polymer chains would entangle and polymerize, causing the aerogel structure to shrink, thereby accelerating the drug release from the aerogel center [15,39]. These results indicate that the prepared drug-loaded aerogel has good temperature-sensitive stimulation properties. As shown in Figure 6b, at a low pH condition, the carboxyl groups on the drug carrier were protonated to form hydrogen bonds, and the aerogels’ structure shrank inward, causing the drug to be squeezed and released. As the pH increased, the carboxyl groups on the drug carrier formed an anionic end group. Due to the mutual repulsion of same charges, the aerogels’ volume will expand outwardly, and the concentration gradient will result in the drug release [40].

#### 3.7.4. Drug Release Kinetics

Figure 7 shows the fitting curves of the sustained-release kinetics of the drug-loaded PNIPAM, CMC/PNIPAM, and CMC/Ca^2+^/PNIPAM aerogels. Various release kinetic models (including first-order kinetic models and Higuchi and Korsmeyer–Peppas models) were used to fit the drug release data of the three 5-FU-loaded aerogels under different temperature and pH conditions. The results demonstrated that the first-order kinetic model and the Korsmeyer–Peppas model were more accurate in fitting the sustained-release curve of the drug. The results of the Korsmeyer–Peppas model demonstrated that the release of the drug was mainly controlled by diffusion, and the extrusion of the drug was accelerated due to the shrinkage of the aerogel structure [15].

Table 3 presents the fitting parameters of the slow-release kinetics of the PNIPAM, CMC/PNIPAM, and CMC/Ca^2+^/PNIPAM aerogels under different temperature and pH conditions. In different release media, the correlation coefficients (R^2^) of the first-order kinetic model and the Higuchi and Korsmeyer-Peppas models of the drug-loaded PNIPAM, CMC/PNIPAM, and CMC/Ca^2+^/PNIPAM aerogels all exceeded 0.92, indicating that the drug sustained-release model can well fit the drug sustained-release curve. In the Korsmeyer–Peppas model, the parameter *n* characterizes the release mechanism. When *n* < 0.45, the sustained-release mechanism of drug-loaded aerogels is mainly Fickian diffusion. When 0.45 < *n* < 0.89, the release mechanism of the drug-loaded aerogel is mainly controlled by non-Fickian diffusion; when *n* is greater than 0.89, the drug release mechanism is mainly affected by the bone erosion effect of the drug-loaded aerogel [41]. The release index *n* (*n* less than 0.45) of the Korsmeyer–Peppas model indicates that the drug releases of the drug-loaded PNIPAM, CMC/PNIPAM, and CMC/Ca^2+^/PNIPAM aerogels were controlled by Fickian diffusion; this is also confirmed by the high correlation of the Higuchi model. The Fickian diffusion mechanism was mainly because 5-FU molecules interacted with the pore surface through the hydrogen bonding effect.

## 4. Conclusions

Temperature/pH-responsive CMC/poly (*N*-isopropyl acrylamide) IPN aerogels were developed as degradable cellulose aerogel-based drug carriers. The aerogels were highly porous and low bulk density, with abundant functional groups and interconnected pores structures, making them ideal drug carriers. The maximum adsorption capacities of the PNIPAM, CMC/PNIPAM, and CMC/Ca^2+^/PNIPAM aerogels for the anticancer drug 5-FU were 87.24 mg/g, 161.33 mg/g, and 199.30 mg/g, respectively, which proves that the three aerogels can be used as effective drug carriers. The temperature and pH responsive properties of aerogels can control the release amount and release rate of the drug under different sustained-release conditions. The sustained-release model of aerogels reveals that the drug release is mainly controlled by Fickian diffusion. Cytotoxicity studies have demonstrated that CMC/Ca^2+^/PNIPAM aerogels have great potential for biomedical applications due to their biodegradability and biocompatibility.

## Figures and Tables

**Figure 1 polymers-14-01578-f001:**
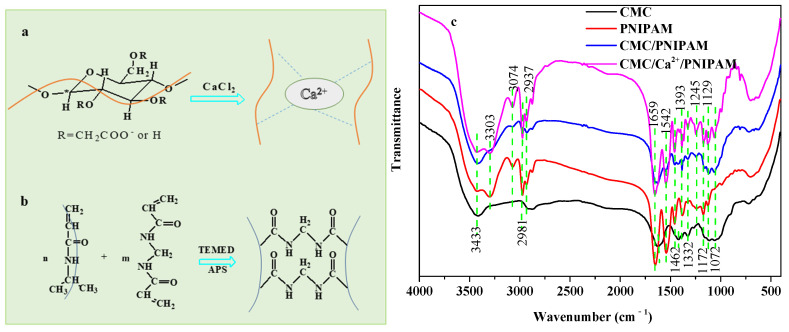
(**a**,**b**) Synthesis of CMC/Ca^2+^/PNIPAM aerogels. (**c**) FTIR spectra of CMC and PNIPAM, CMC/PNIPAM, and CMC/Ca^2+^/PNIPAM aerogels.

**Figure 2 polymers-14-01578-f002:**
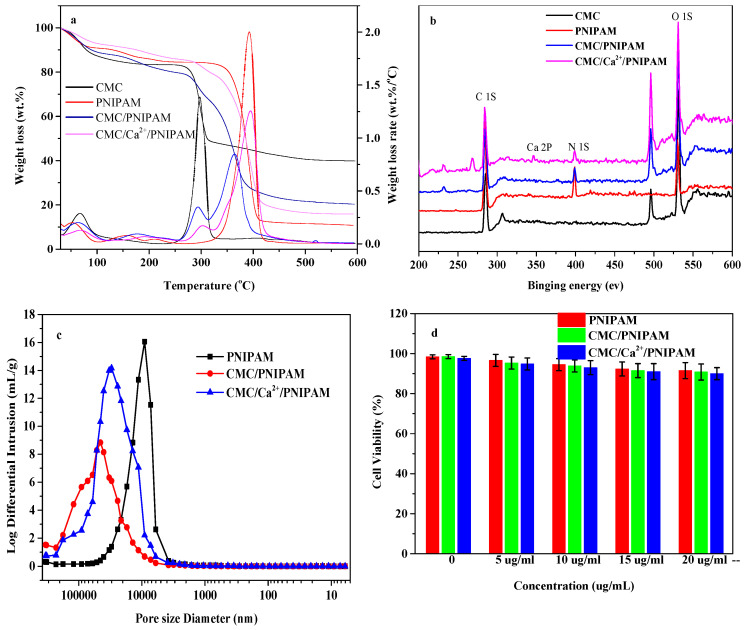
(**a**) Weight loss and weight loss rate of CMC and aerogels; (**b**) XPS patterns of CMC and aerogels; (**c**) pore size distribution curve of aerogel; (**d**) cytotoxicity analysis of aerogel.

**Figure 3 polymers-14-01578-f003:**
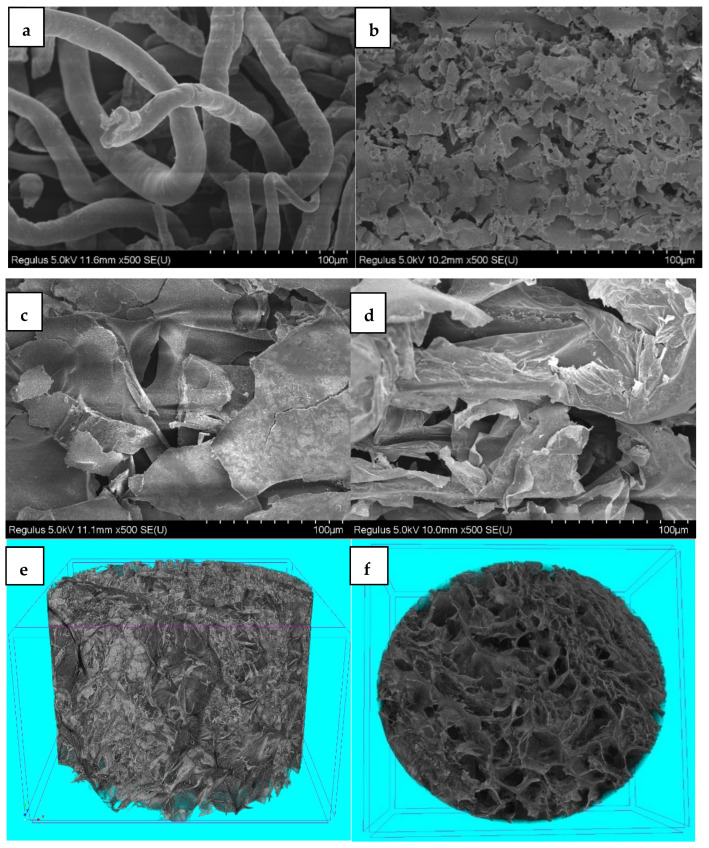
SEM images of (**a**) CMC and (**b**) PNIPAM, (**c**) CMC/PNIPAM, and (**d**) CMC/Ca^2+^/PNIPAM aerogels; Micro-CT stereogram (**e**) and cross section (**f**) of CMC/Ca^2+^/PNIPAM aerogels.

**Figure 4 polymers-14-01578-f004:**
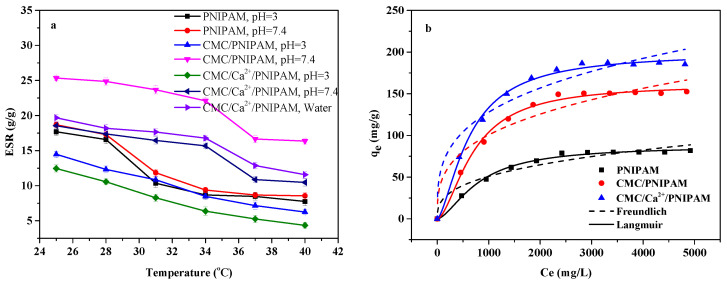
(**a**) ESRs of aerogels under different conditions; (**b**) isothermal adsorption curves of aerogels.

**Figure 5 polymers-14-01578-f005:**
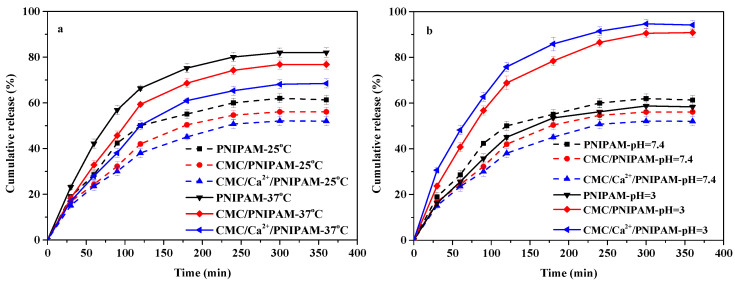
(**a**) Effect of temperature on drug release; (**b**) effect of pH on drug release.

**Figure 6 polymers-14-01578-f006:**
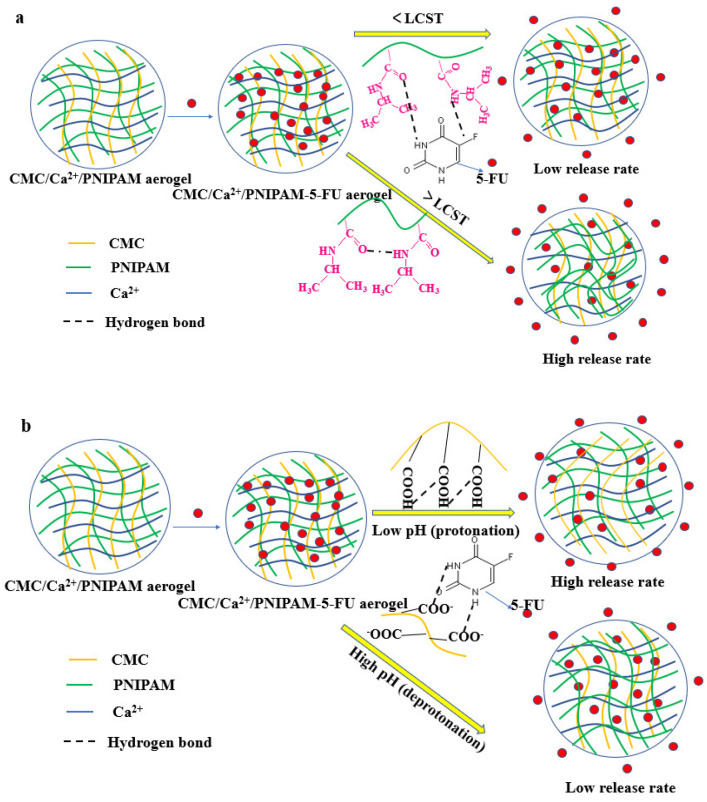
(**a**) Release process for aerogels under different temperatures; (**b**) release process for aerogels under different pH conditions.

**Figure 7 polymers-14-01578-f007:**
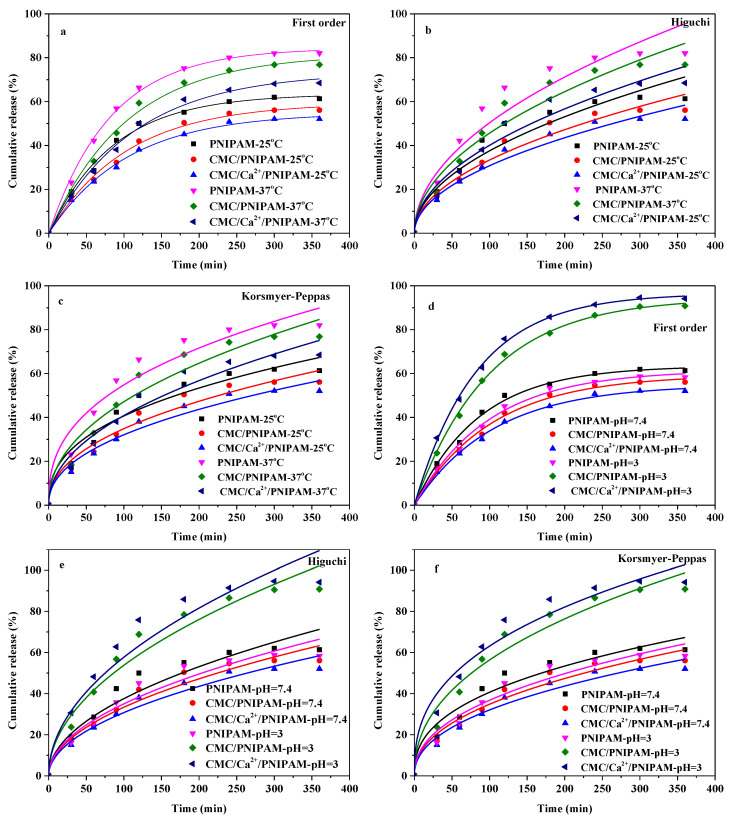
Fitting curve of drug release kinetics of aerogels. (**a**–**c**) are the First-Order, Higuchi and Korsmeyer-Peppas model of aerogels at different temperatures, respectively; (**d**–**f**) are the First-Order, Higuchi and Korsmeyer–Peppas model of aerogels at different pH.)

**Table 1 polymers-14-01578-t001:** Properties of CMC and aerogels.

Samples	C	N	O	H	Density g/cm^3^	Porosity %	Charge Density mmol/g
CMC	35.52	0.00	45.19	5.44	--	--	−4.96
PNIPAM	56.07	12.34	22.39	9.54	0.086	92.6	−0.16
CMC/PNIPAM	42.58	6.59	33.29	7.22	0.079	93.8	−1.08
CMC/Ca^2+^/PNIPAM	37.50	6.19	40.69	6.11	0.072	94.7	−1.92

**Table 2 polymers-14-01578-t002:** Isothermal adsorption curve constants of 5-FU-loaded aerogels.

Samples	Langmuir Model			Freundlich Model		
	*q*_*m*_ (mg/g)	*K*_*L*_ (L/mg)	R^2^	1/*n*	*K*_*F*_ (L/mg)	R^2^
PNIPAM	87.24	0.0009	0.9844	0.3484	4.5895	0.8513
CMC/PNIPAM	161.33	0.0011	0.9835	0.3188	11.1312	0.8236
CMC/Ca^2+^/PNIPAM	199.30	0.0013	0.9857	0.2948	16.6911	0.8382

**Table 3 polymers-14-01578-t003:** Release kinetics for 5-FU-loaded aerogels under different conditions.

Reaction Conditions	First-Order			Higuchi		Korsmeyer-Peppas		
	Q*_max_* (%)	*K* (min^−1^)	R^2^	a	R^2^	*K* (min^−*n*^)	*n*	R^2^
1, 25 °C, 7.4	63.46	0.01165	0.9941	0.0376	0.9314	0.0665	0.3933	0.9484
2, 25 °C, 7.4	59.37	0.00967	0.9933	0.0334	0.9595	0.0454	0.4425	0.9606
3, 25 °C, 7.4	54.91	0.00956	0.9965	0.0308	0.9674	0.0412	0.4454	0.9689
1, 37 °C, 7.4	84.39	0.01207	0.9977	0.0504	0.9249	0.0925	0.3862	0.9452
2, 37 °C, 7.4	81.62	0.00954	0.9935	0.0456	0.9502	0.0574	0.4473	0.9467
3, 37 °C, 7.4	73.39	0.00885	0.9942	0.0401	0.9605	0.0456	0.4460	0.9560
1, 25 °C, 3	61.52	0.01012	0.9948	0.0350	0.9496	0.0496	0.4349	0.9515
2, 25 °C, 3	94.52	0.01007	0.9981	0.0538	0.9585	0.0751	0.4376	0.9606
3, 25 °C, 3	96.54	0.0121	0.9985	0.0577	0.9336	0.1100	0.3793	0.9597

Note: PNIPAM aerogel: 1, CMC/PNIPAM aerogel: 2, and CMC/Ca^2+^/PNIPAM aerogel: 3.

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
