# Peer review of "Temperature/pH-Responsive Carboxymethyl Cellulose/Poly (N-isopropyl acrylamide) Interpenetrating Polymer Network Aerogels for Drug Delivery Systems"

_polymers, 2022, doi:10.3390/polym14081578_

Round 1
Reviewer 1 Report
It is quite interesting paper, the results are new and sound. It would be nice to discuss in the text the Molecular dynamics simulations of drug release kinetics from polymeric matrices and kinetics polymer swelling. It would be nice to understand whether the auuthors semm some prospects to apply the MD to investigate the discuussed in paper spolymeric systems. In this resppect, I would recommend following new quite new references to discuss them in the text.
- Gurina D.L. et al., Journal of Molecular Liquids. 2022. Article 118758
- Gurina D., et al. Journal of Physical Chemistry B. 2020. Vol. 124. No. 38. P. 8410-8417.
Author Response
Response: Thank you for your comment. In the manuscript, two drug adsorption models and three drug sustained release kinetic models were used to fit the loading and release of the drug, and the related mechanism of drug loading and sustained release was explored. In the next article, we will conduct a special study on the molecular dynamics simulation of the swelling kinetics of aerogels in different environments, which will not be described in this article. Thank you for your understanding.
Reviewer 2 Report
- Please check 2.3 and 2.4, they are the same.
- Please mention the number of replications for analytical methods.
Author Response
1.Please check 2.3 and 2.4, they are the same.
Response: Thank you for your comment. 2.3 and 2.4 have been revised in the manuscript.
2.Please mention the number of replications for analytical methods.
Response: Thank you for your comment. The number of replicates for the analytical method has been added in the manuscript.
Reviewer 3 Report
Manuscript titled as “Temperature/pH-responsive carboxymethyl cellulose/poly(N-isopropyl acrylamide) interpenetrating polymer network aerogels for drug delivery systems” describes new and interesting results including synthesis and application of the temperature/pH-responsive carboxymethyl cellulose/poly (N-isopropyl acrylamide) interpenetrating polymer network aerogels. This paper can be published in Polymers mdpi after major revision.
First and the main question, what new are presented in this manuscript if compare with similar paper "Temperature-responsive hydroxypropyl methylcellulose-N-isopropylacrylamide aerogels for drug delivery systems" that was published previously.
I think, fig. 1a is difficult for understanding, please schematically depict all stages of the synthesis, add conditions of the synthesis and show the final product in a more suitable manner.
Appropriate discussion on temperature/pH-responsive properties is absent in the manuscript. What molecular mechanisms of responsivity were realized? Is it interaction between carboxylic groups and amide fragments? An appropriate scheme should be added.
Information about the swelling behavior of the interpenetrating polymer network aerogels in "pure" water should be added. What about the impact of the Ca ions on LCST, my previous experience suggests possible shifts in LCST.
What buffer solutions were used in measurements? Please add their formulation. Appropriate discussion about the impact of the buffer solution on temperature-responsive properties of the sensitive polymers is absent.
Interactions between amide units of the PNIPAM at a temperature above LCST and PNIPAM and water are incorrect (Figure 6). Interaction between carboxylic groups and amide units at low pH is absent. Discussion about the interaction of the 5-fluorouracil with functional groups of CMC/Ca2+/PNIPAM aerogels via physical embedment and hydrogen bonding interactions are almost absent. Figure 6 should be essentially improved.
I would like to recommend to cite the following references where impact of the pH on temperature-responsive systems was presented.
https://doi.org/10.1007/s00396-020-04750-0
https://doi.org/10.1007/s00396-022-04959-1
https://doi.org/10.1038/373049a0
Author Response
- First and the main question, what new are presented in this manuscript if compare with similar paper "Temperature-responsive hydroxypropyl methylcellulose-N-isopropylacrylamide aerogels for drug delivery systems" that was published previously.
Response: Thank you for your comment. This article endows aerogels with temperature and pH responsive properties, and simultaneously analyzes the pore structure and cytotoxicity of aerogels. This article also adds the mechanistic analysis of aerogel loading and sustained drug release, which are not mentioned in the "Temperature-responsive hydroxypropyl methylcellulose-N-isopropylacrylamide aerogels for drug delivery systems". Compared with the previous article, the aerogel in this article has significantly increased drug loading and prolonged drug release time.
- I think, fig. 1a is difficult for understanding, please schematically depict all stages of the synthesis, add conditions of the synthesis and show the final product in a more suitable manner.
Response: Thank you for your comment. The reaction scheme of Figure 1a has been redrawn in the manuscript.
- Appropriate discussion on temperature/pH-responsive properties is absent in the manuscript. What molecular mechanisms of responsivity were realized? Is it interaction between carboxylic groups and amide fragments? An appropriate scheme should be added.
Response: Thank you for your comment. Molecular structure transition of temperature/pH-responsive aerogels from coils to spheres in aqueous solution via multiple mechanisms. Its response performance is mainly reflected in the change of swelling performance. Under the conditions of lower than LCST and high pH, the amide group of PNIPAN and the carboxyl group of CMC polymers in the aerogel form hydrogen bonds with water molecules and are in a hydrophilic state, and the swelling performance of the aerogel is high. In contrast, at conditions above the LCST and at low pH, the hydrogen bonds between the polymer and water molecules are disrupted, and the conformation of the aero-gel polymer changes from a loose coil to a folded chain, which is characteristic of the hydrophobic state and more low swelling properties. Hydrogen bonds between carboxyl and amide fragments are formed in the interpenetrating network CMC/Ca2+/PNIPAM aerogel, which will improve the performance of the aerogel.
- Information about the swelling behavior of the interpenetrating polymer network aerogels in "pure" water should be added. What about the impact of the Ca ions on LCST, my previous experience suggests possible shifts in LCST.
Response: Thank you for your comment. The swelling behavior of the interpenetrating polymer network aerogels in "pure" water has been added in the manuscript. The introduction of calcium ions leads to a slight decrease in the LCST temperature of the aerogel, which is consistent with the effect of salt ions on the critical temperature in the literature. The above content has been added in the manuscript.
- What buffer solutions were used in measurements? Please add their formulation. Appropriate discussion about the impact of the buffer solution on temperature-responsive properties of the sensitive polymers is absent.
Response: Thank you for your comment. The types and main components of the buffers have been added in the Materials section of the manuscript. The LCST temperature of aerogels in acidic PBS buffer solutions is slightly lower than that in neutral PBS buffer conditions. The above content has been added in the manuscript.
- Interactions between amide units of the PNIPAM at a temperature above LCST and PNIPAM and water are incorrect (Figure 6). Interaction between carboxylic groups and amide units at low pH is absent. Discussion about the interaction of the 5-fluorouracil with functional groups of CMC/Ca2+/PNIPAM aerogels via physical embedment and hydrogen bonding interactions are almost absent. Figure 6 should be essentially improved.
Response: Thank you for your comment. Amide bonds and hydrogen bonding of carboxyl groups to the drug have been added in Figure 6, and the drug loading the drug through hydrogen bonding and physical entrapment content has been added in the manuscript.
- I would like to recommend to cite the following references where impact of the pH on temperature-responsive systems was presented.
https://doi.org/10.1007/s00396-020-04750-0
https://doi.org/10.1007/s00396-022-04959-1
https://doi.org/10.1038/373049a0
Response: Thank you for your comment. The above three references have been read carefully and cited in the manuscript.
Round 2
Reviewer 1 Report
Manuscript can be published in its present form.
Reviewer 3 Report
The quality of the manuscript was essentially improved after revision. The manuscript can be accepted for publication in its present form.